# VBQ-Net: A Novel Vectorization-Based Boost Quantized Network Model for Maximizing the Security Level of IoT System to Prevent Intrusions

Ganeshkumar Perumal [1], Gopalakrishnan Subburayalu [2], Qaisar Abbas [1,*], Syed Muhammad Naqi [3] and Imran Qureshi [1]

1   College of Computer and Information Sciences, Imam Mohammad Ibn Saud Islamic University (IMSIU), Riyadh 11432, Saudi Arabia; gpperumal@imamu.edu.sa (G.P.); iqureshi@imamu.edu.sa (I.Q.)
2   Department of Information Technology, Hindustan Institute of Technology and Science, Kelambakkam, Chennai 603103, India
3   Department of Computer Science, Quaid-i-Azam University, Islamabad 44000, Pakistan; smnaqi@qau.edu.pk
*   Correspondence: qaabbas@immau.edu.sa

**Abstract:** Data sharing with additional devices across wireless networks is made simple and advantageous by the Internet of Things (IoT), an emerging technology. However, IoT systems are more susceptible to cyberattacks because of their continued growth and technological advances, which could lead to powerful assaults. An intrusion detection system is one of the key defense mechanisms for information and communications technology. The primary shortcomings that plague current IoT security frameworks are their inability to detect intrusions properly, their substantial latency, and their prolonged processing time and delay. Therefore, this work develops a clever and innovative security architecture called Vectorization-Based Boost Quantized Network (VBQ-Net) for protecting IoT networks. Here, a Vector Space Bag of Words (VSBW) methodology is used to reduce the dimensionality of features and identify a key characteristic from the featured data. In addition, a brand-new classification technique, called Boosted Variance Quantization Neural Networks (BVQNNs), is used to classify the different types of intrusions using a weighted feature matrix. A Multi-Hunting Reptile Search Optimization (MH-RSO) algorithm is employed during categorization to calculate the probability value for selecting the right choices while anticipating intrusions. In this study, the most well-known and current datasets, such as IoTID-20, IoT-23, and CIDDS-001, are used to validate and evaluate the effectiveness of the proposed methodology. By evaluating the proposed approach on standard IoT datasets, the study seeks to address the limitations of current IoT security frameworks and provide a more effective defense mechanism against cyberattacks on IoT systems.

**Keywords:** internet of things (IoT); intrusion detection system (IDS); vectorization-based boost quantized network (VBQ-Net); vector space bag of words (VSBW); boosted variance quantization neural network (BVQNN); multi-hunting reptile search optimization (MH-RSO)



## 1. Introduction

The Internet of Things (IoT) is a new technology that enables the seamless linking of various devices to deliver services and automate processes across a variety of industries [1,2], from our everyday lives to vital infrastructure systems. A new statistical analysis estimates that the worldwide IoT market was approximately USD 296 billion in 2020 and will be USD 1580 billion in 2025. IoT devices have become increasingly common in recent years, and IoT networks are crucial to both industry and everyday life [3–5]. They simplify each element of everyday life, but they are also susceptible to different attacks, which somewhat offsets their convenience-improving effects. However, the IoT's [6] quick spread to almost every aspect of life has given rise to a number of new cybersecurity threats. This is due to the fact that IoT devices frequently have low computing and energy capabilities,

leaving them more vulnerable to attackers. Unfortunately, IoT devices are more vulnerable and difficult to defend against cyberattacks than computers. Therefore, identifying attacks to shield IoT devices against unlawful behavior is essential for expanding the IoT's applications [7,8]. Moreover, the attack detection methodologies are categorized into signature-based and intelligence-based methods. Jamming is one of the classes of DoS attacks that disrupt normal network communication by creating interference among IoT devices [9,10]. These attacks harm the network because they have a detrimental impact on the functionality of small devices with limited resources. An adversary's goal in these attacks is often to stifle communication, in order to drain the limited energy reserves of small devices. Jamming is one of the most damaging attacks that can block wireless communication channels by disabling radio communication bands in networks by sending fake packets. As a result, it poses a serious risk to IoT networks [11,12], which are made up of nodes with scarce resources and power. Jamming can be done continuously or sporadically.

An essential kind of assault on the physical and medium access control layers is reactive jamming, which results in the unnecessary use of energy by tiny devices during communication [13]. Security attack modeling allows us to decide on mitigation measures and understand how a jamming attack actually occurs in IoT networks. To handle jamming threats in the IoT arena, including reactive jamming, numerous protective measures utilizing a variety of defense mechanisms have been developed. Similarly, a Distributed Denial-of-Service (DDoS) attack is one of the most popular assaults that attempts to block a device's lawful functions and render the device inaccessible to authorized users by using excessive amounts of attack resources [14]. Moreover, this attack uses a distributed attack source to hinder or prevent authorized users from accessing a particular network service or resource. Typically, signature-based techniques look for IoT attack signatures in the incoming network traffic. To defeat the signatures using these methods, one needs to have a thorough understanding of known IoT assaults [15,16]. On the other hand, machine learning-based approaches make an effort to understand the characteristics of legitimate and harmful data during the training and testing phases. These models are used to identify assaults in the incoming traffic during the prediction phase. Machine-learning-based [17,18] solutions can quickly identify different IoT assaults, since they have the potential to learn essential information and features automatically and gradually from acquired data. In this research work, different types of security threats in IoT networks are detected and classified to ensure the safety of data communication.

The rapid growth of Internet of Things (IoT) devices has ushered in a new era of interconnectedness and convenience. However, the increasing deployment of IoT systems has also led to significant security challenges, with hazardous intrusions posing a severe threat to the integrity and privacy of sensitive data. To address this issue, we propose a cutting-edge solution: a novel Vectorization-Based Boost Quantized Network Model (VBQNM) tailored explicitly for safeguarding IoT systems from malicious intrusions.

### 1.1. Major Contributions

The VBQNM leverages the power of vectorization and boosting techniques to enhance the robustness of IoT security. By transforming data into vectorized representations, the model can efficiently process and analyze incoming information, thereby enabling faster detection of and response times to potential intrusions. Additionally, the utilization of boosting algorithms enables the model to dynamically adapt and improve its accuracy over time, keeping up with emerging attack strategies. The following are the main contributions:

○ In this work, an intelligent and novel security framework, named Vectorization-Based Boost Quantized Network (VBQ-Net), is developed for securing IoT systems;

○ To select essential attributes from the featured data, a Vector Space Bag of Words (VSBW) methodology is implemented, which minimizes the dimensionality of features;

○ Also, a new Boosted Variance Quantization Neural Network (BVQNN)-based classification algorithm is used with the weighted feature matrix to classify the type of intrusion from the given data;

○ During classification, a Multi-Hunting Reptile Search Optimization (MH-RSO) algorithm is used to optimally compute the probability value for making proper decisions while predicting intrusions;

○ The most popular recent datasets, such as IoTID-20, IoT-23, and CIDDS-001, are used in this study for validating and assessing the performance of the suggested framework.

*1.2. Paper Organization*

The remaining portions of this article are divided into the following sections: An in-depth literature study on the predominant IoT threat detection methods used in conventional work is presented in Section 2. The operating model and comprehensive description of the proposed VBQ-Net architecture are then described in Section 3. Additionally, Section 4 analyzes a range of traits and datasets to assess the consequences and efficacy of the suggested framework. Discussions are described in Section 5. The conclusions and recommendations in Section 6 bring the entire text to a close.

## 2. Literature Review

This section presents a thorough, in-depth overview of the various machine learning and deep learning techniques utilized in the literature for IoT security.

Fadele et al. [19] implemented a new countermeasure technique for spotting jamming attacks in the IoT. The suggested methodology is developed based on the consistency algorithm, where the threshold value is estimated to maximize the attack detection accuracy. The authors of this paper mainly focused on identifying jamming attacks for reliable and secure packet transmission. Jia et al. [20] introduced a new defense mechanism named FlowGuard for protecting IoT networks from DDoS attacks. The purpose of this study is to identify and detect the different types of DDoS attacks in the network, which include flooding-based, naive flooding, protocol exploitation, and reflection-based attacks. Vu et al. [21] deployed a deep transfer learning technique for spotting attacks from IoT networks. Here, one of the most effective learning algorithms called the auto-encoder methodology, is used for attack identification and classification. In this method, the loss function, also termed a reconstruction error, is estimated according to the input and output data. Yet, the suggested model requires more time for detection, which degrades the efficacy of the overall prediction system.

AI-Othman et al. [22] investigated some of the recent state-of-the-art methodologies for the detection and classification of botnet attacks. The majority of IoT botnet detection methods use a behavioral strategy that relies primarily on artificial intelligence, with some cutting-edge studies using deep learning. Modern solutions do not often use hybrid or signature-based methods. The study indicates that the accuracy rate was generally seen as the primary performance metric of the selected classification models. Hekmati et al. [23] compared the performance of four different classification techniques, such as multilayer perceptron (MLP), CNN, auto-encoder, and long-short-term memory (LSTM), for choosing the most reliable mechanism for increasing the security of IoT networks. In this study, the different types of cyber-attack datasets used for IoT networks are examined, which include DARPA, CAIDA, CICIDS 2017, and BoT-IoT. Moreover, the most common performance measures, such as accuracy, precision, and recall, are used for validating the training and testing results of these four methods. Then, the analytical results of this study indicated that the LSTM outperformed the other three classifiers with superior results.

Khraisat et al. [24] implemented an ensemble classification approach for spotting intrusions from IoT networks. Typically, three different types of intrusion detection methodologies were used in the previous studies: Hybrid IDS, Signature IDS, and anomaly-based IDS. The main purpose of this article is to develop an effective framework for detecting intrusions with increased accuracy and a low false alarm rate. Qiu et al. [25] introduced a deep-learning-based IDS for detecting new adversarial attacks from IoT networks. Here, the Kitsune-based intrusion detection system is developed for categorizing benign and malignant traffic from the network. In this module, the feature extractor and feature mapper

are used to estimate the temporal statistics of the data packets in the stream. Based on this value, normal and anomalous network traffic are categorized in the network. In order to validate the system, different types of performance measures such as false positives, false negatives, accuracy, and root mean square value have been estimated. According to this value, the performance of the attack detection system is determined.

Hajiheidari et al. [26] presented a comprehensive analysis to examine the different types of attack detection methodologies used in IoT systems. Here, intrusion detection based on location, evaluation techniques, and attacks is reviewed and investigated. Ferrag et al. [27] introduced a rule-based decision tree classification mechanism for safeguarding IoT networks. The authors intend to reduce the false acceptance rate and improve the detection rate of the suggested system using a rule-based classification algorithm. This article's [22] main objective is to identify botnet attacks using cutting-edge detection techniques. The bulk of IoT botnet detection techniques employ a behavioral approach, which is mostly based on machine learning, with some state-of-the-art research utilizing deep learning. Hybrid or signature-based strategies are not widely used in modern solutions. Additionally, this study suggests using a hybrid approach to detect IoT botnets, because a hybrid IDS that combines signature-based and behavioral detection may have a higher success rate at finding IoT botnets. In such a system, signature-based detection would help in quickly recognizing previously experienced attacks, while the behavioral detection component would handle any remaining zero-day or unrecognizable assaults.

Kasongo et al. [28] designed a machine learning algorithm for detecting intrusions from the UNSW-NB 15 dataset. In this study, the most standard machine learning algorithms, such as Support Vector Machine (SVM), Decision Tree (DT), Artificial Neural Network (ANN), K Nearest Neighbor (KNN), and Logistic Regression (LR), have been validated and compared to choose the most reliable technique for intrusion detection. One of the best and most adaptive ML models, SVM, can perform tasks such as regression and classification. The goal is to divide the data set into distinct classes in an effort to identify the best hyperplanes. One of the advantages of using SVM is that it frequently functions well for high-dimensional source fields. Additionally, the SVM model enables the decision-making process to choose among an array of Kernel functions. Despite being called "regression", the ML technique known as LR is generally used for binary classification tasks. In multiclass classification jobs where the learning algorithm uses one-vs.-rest methods, the LR can also be applied. The sigmoid function or one of its variants is applied to a linear ML model in the LR model. This operation results in a compressed output between [0,1]. With the aid of the UNSW dataset, the authors in [29] employed a machine learning technique to identify botnet attacks. According to the criteria of testing accuracy, precision, and detection rate, the performance of four distinct machine learning techniques—KNN, SVM, NB, and DT—was evaluated in this paper. Additionally, the Principal Component Analysis (PCA) method is utilized to shrink the dimension of the dataset, in order to streamline the classification process.

According to this literature review, as mentioned in Table 1, the major drawbacks behind the existing techniques have been identified, which include increased delay, lower detection performance, high energy consumption, and longer processing times. In order to resolve these issues, an efficient intrusion detection approach is developed in this research work for maximizing the security level of IoT networks.

**Table 1.** The table contains a literature review of several research works related to intrusion detection and attack classification in Internet of Things (IoT) networks.

| Cited | Methodology | Limitations |
|---|---|---|
| Fadele et al. [19] | Implemented a countermeasure technique based on the consistency algorithm to detect jamming attacks in IoT. They focused on reliable and secure packet transmission. | Lack of information on the actual effectiveness in real IoT environments. |
| Jia et al. [20] | Introduced FlowGuard, a defense mechanism for protecting IoT networks from DDoS attacks. They aimed to detect different types of DDoS attacks in the network. | May not cover all possible DDoS attack variations. |
| Vu et al. [21] | Deployed a deep transfer learning technique using auto-encoders to spot attacks in IoT networks. However, the suggested model was time-consuming for detection. | The suggested model's time-consuming nature affects detection efficacy. |
| AI-Othman et al. [22]: | Investigated state-of-the-art methodologies for detecting and classifying botnet attacks in the IoT. They highlighted the importance of accuracy as the primary performance metric for classification models. | Limited exploration of hybrid or signature-based detection methods. |
| Hekmati et al. [23]: | Compared four classification techniques (MLP, CNN, auto-encoder, LSTM) to improve IoT network security, with LSTM outperforming others. | Evaluation limited to specific classifiers, may not generalize to other datasets. |
| Khraisat et al. [24]: | Implemented an ensemble classification approach to detect intrusions in IoT networks, aiming for increased accuracy and a low false alarm rate. | May not provide insights into the performance of individual intrusion detection methods within the ensemble. |
| Qiu et al. [25]: | Introduced a Kitsune-based intrusion detection system using deep learning to categorize benign and malignant traffic in IoT networks. | Performance evaluation metrics used may not cover all aspects of attack detection. |
| Hajiheidari et al. [26]: | Presented a comprehensive analysis of different attack detection methodologies used in IoT systems. | Lack of quantitative results on the effectiveness of reviewed attack detection methodologies. |
| Ferrag et al. [27]: | Developed a rule-based decision tree classification mechanism to reduce false acceptance rate and improve detection rate for IoT networks. | Evaluation might not be performed on a diverse set of IoT attack scenarios. |
| Kasongo et al. [28]: | Designed a machine learning algorithm for detecting intrusions using the UNSW-NB 15 dataset. They compared various ML algorithms (SVM, DT, ANN, KNN, LR) and used PCA for dimensionality reduction. | Performance comparison limited to specific ML algorithms and dataset (UNSW-NB 15). |
| Proposed | Developed VBQ-Net, a security framework for IoT systems that uses VSBW to select essential data attributes and BVQNN for classification. MH-RSO algorithm optimizes probability values. Validated the framework using popular datasets like IoTID-20, IoT-23, and CIDDS-001. | It is necessary to keep VBQ-Net updated and improved to stay protected from new intrusion methods. |

## 3. Materials and Methods

*Proposed Framework*

This section gives a thorough discussion of the suggested security paradigm that would be used to safeguard IoT networks. The main thing that this research adds is a new security system called Vectorization-Based Boost Quantized Network (VBQ-Net) that can

protect IoT systems from potentially dangerous intrusions with a higher rate of detection and a lower rate of wrong predictions. The proposed system's workflow model is depicted in Figure 1, which incorporates the following safety operation modules:

○     Network-featured data collection;
○     Feature selection using the Vector Space Bag of Words (VSBW);
○     Intrusion identification using a Boosted Variance Quantization Neural Network (BVQNN);
○     Probability estimation using Multi-Hunting Reptile Search Optimization (MH-RSO);
○     Performance evaluation.

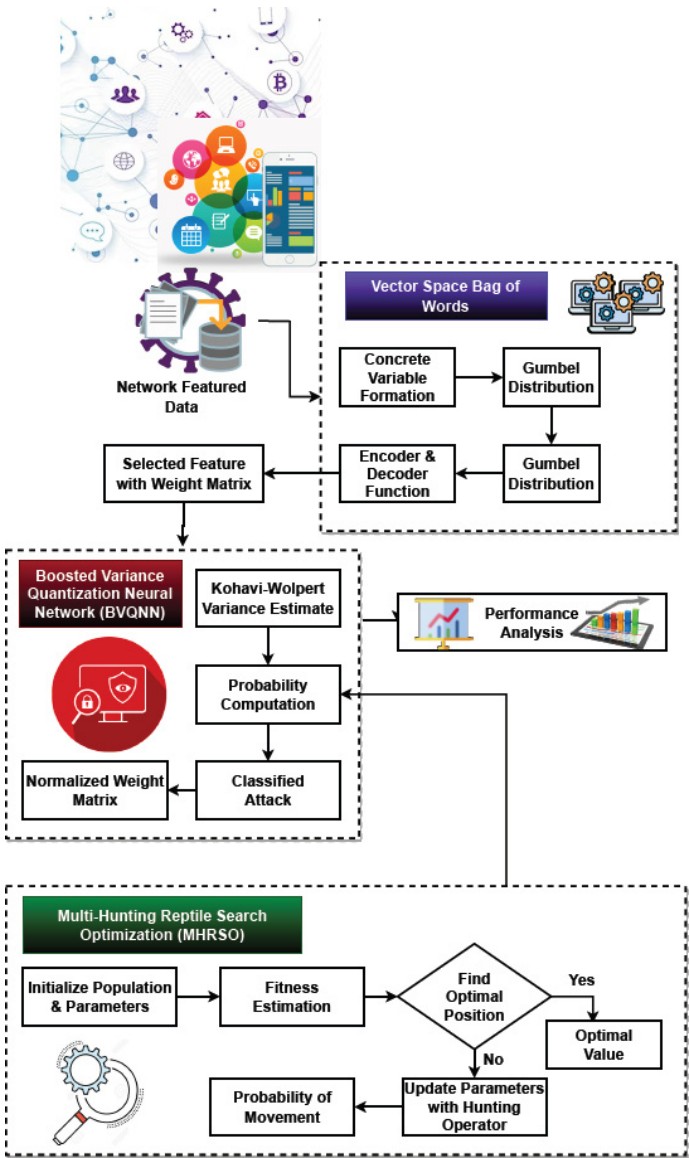

**Figure 1.** A systematic flow diagram of the proposed system.

For system deployment and validation, network-featured data, such as IoT-23 [30], IoT-ID 20 [31], and CIDDS-001 [32], are initially collected. The IoT-23 dataset has 34,106 instances of normal and malicious traffic from IoT devices. It helps evaluate intrusion detection techniques. The IoT-ID 20 dataset has 18,441 instances of normal and attack traffic, focusing on differentiating between benign and malicious behavior. The CIDDS-001 dataset has 13,042 instances of normal traffic and attacks targeting IoT and industrial control networks. It evaluates intrusion detection systems in IoT and ICS settings. These datasets have various attack types and normal activities, making them great for evaluating VBQ-

Net's intrusion detection abilities. They cover different attack categories and real-world scenarios, making them valuable resources for testing.

By using the required feature matrix of the resulting dataset, the Vector Space Bag of Words (VSBW) technique is applied to choose the pertinent features. The VSBW technique is employed to reduce feature dimensionality and extract key characteristics from the network traffic data, facilitating effective intrusion detection and categorization within IoT networks. The VSBW methodology is tailored to capture the unique features and patterns present in network data traffic, which includes attributes such as packet sizes, inter-arrival times, source and destination IP addresses, and port numbers, among others. The following processes are carried out during feature selection: concrete variable formation, Gumbel distribution, encoding, and decoding. Based on the weighted feature matrix, the innovative Boosted Variance Quantization Neural Network (BVQNN) technique is used to properly predict and categorize the type of intrusion. During this process, Kohavi–Wolpert variance estimation, probability calculation, and normalized weight matrix generation are carried out. The Multi-Hunting Reptile Search Optimization (MHRSO) technique is employed to optimally compute the probability value in order to make decisions while anticipating the type of intrusion. Finally, a number of measurements are used to evaluate the performance of the projected label.

The training process involves a standard data-splitting procedure to create training and testing subsets. The dataset is randomly divided into two parts: a training set and a testing set. The typical ratio used is 70% for training and 30% for testing, ensuring a balance between model training and evaluation. The proposed VBQ-Net model uses Boosted Variance Quantization Neural Networks to classify intrusion types based on pre-processed network traffic data. During training, hyperparameters and components are fine-tuned using techniques like cross-validation to achieve optimal performance. The model's effectiveness is evaluated using relevant metrics and compared to existing intrusion detection techniques to determine its superiority.

A    **Vector Space Bag-of-Words (VSBW) for Feature Selection:** The VSBW technique is used to select the most necessary and pertinent features for identifying and categorizing the type of intrusion from the IoT networks after the input dataset has been collected. Overall steps of VSBW are described in Algorithm 1. The dataset properties typically have a significant impact on intrusion detection performance, since they aid in the analysis of attack characteristics. Some optimization strategies were used in the earlier works for feature selection and dimensionality reduction. These methods struggle with the main issues of long search times, low efficiency, and intricate mathematical modeling. Therefore, the proposed study seeks to employ a straightforward as well as cutting-edge methodology for feature selection. In the proposed VSNW model, the following operations are performed to choose the relevant features:

(a)    Concrete variable formation;
(b)    Gumbel distribution;
(c)    Encoding and decoding operations;
(d)    Weight matrix formation.

The main purpose of using this technique is to reduce the model size of the dataset by choosing the features with concrete variables. Despite the fact that the selection of features problem is typically a combinatorial one, because it disrupts gradient propagation and makes complex end-to-end training, this problem can be solved by utilizing a concrete random variable for training, which estimates the gradients through the use of the re-parameterization approach and is a perpetual relaxation of a finite set of single-hot vectors. The major advantages of using this technique are increased efficiency, lower time consumption, and reduced dimensionality. After obtaining the featured data, the features of the input can be expressed as shown in the following form:

$$\vartheta_h = \mathcal{O}_h^T * \mathcal{F} \tag{1}$$

where $\mathcal{O}_h \in R^d$ indicates the one-hot vector whose hth feature is 1&0 otherwise. Then, the concrete variables are formed for nth neuron in the selection layer as shown in below:

$$\delta_{(n)} = \frac{\exp^{((\log(\vartheta_h^{(n)} + \mathfrak{g}))/\gamma)}}{\sum_{k=1}^{d} \exp^{((\log(\vartheta_h^{(n)}[k] + \mathfrak{g}_k))/\gamma)}}, k = 1, 2, \ldots, N \tag{2}$$

where $\mathfrak{g} \in R^d$ is derived from the Gumbel distribution, $\gamma$ controlling parameter, and N is the number of selected features. Then, the matrix multiplication process is performed for feature selection as represented in the following equation:

$$F_{sel}(n) = \left\{ \delta_{(1)}, \delta_{(2)}, \ldots \delta_{(N)} \right\}^T \tag{3}$$

Moreover, the low dimensional hidden representation is obtained with the use of encoding as shown in below:

$$\alpha_{enc}^{\Psi}(F_{sel}) = \partial\left[\varpi_{e_L} * \mathcal{F}_h\right] \forall \partial[\varpi_{e_1}, \varpi_{e_2}, \ldots, \varpi_{e_L}] \tag{4}$$

where $\varpi_{e_L}$ indicates weight matrix at each encoder layers, and $e_L$ is the number of encoder layer. By using the softmax layer, the decisions are made as indicated in the following model:

$$\zeta^h(F_{sel}) = softmax\left(\partial\left[\alpha_{enc}^{\Psi}\left(F_{sel}^h\right)\right]\right) \tag{5}$$

where $softmax(.)$ represents the softmax function. Furthermore, the decoding function $\alpha_{dec}^{\Psi}(F_{sel})$ is estimated to compute another hidden representation, $h \in Rh'$, that indicates the last reconstruction layer as shown in below:

$$\alpha_{dec}^{\Psi}(F_{sel}) = \partial\left[\varpi_{D_L} * \mathcal{F}_h\right] \forall \partial[\varpi_{D_1}, \varpi_{D_2}, \ldots, \varpi_{D_L}] \tag{6}$$

where $\varpi_{e_L}$ is the weight matrix at each encoder layers, and $e_L$ is the number of encoder layers. From the decoding result, the final set of selected features are obtained with the weight matrix as shown in below:

$$\mathcal{S}_{\mathcal{F}} = \alpha_{dec}^{\Psi}(F_{sel}) * \varpi_{\mathcal{F}} \tag{7}$$

where $\varpi_{D_L}$ indicates the weight matrix of the feature data. The selected feature $\mathcal{S}_{\mathcal{F}}$ is used by the classifier for predicting intrusions from the dataset.

---

**Algorithm 1: Vector Space Bag-of-Words**

---

Input: Features data $\mathcal{F}_h$
Output: Selected feature $\mathcal{S}_{\mathcal{F}}$

- hth features of the input $\mathcal{F}$ can be expressed as
  $\vartheta_h = \mathcal{O}_h^T * \mathcal{F}$
  where
  $\mathcal{O}_h \in R^d$ indicates the one-hot vector whose hth feature is 1&0 otherwise;
- In the selection layer, concrete variables $\delta_{(n)}$ are formed for the nth neuron with Gumbel distribution $\mathfrak{g} \in R^d$ as shown in Equation (2);
- Perform matrix multiplication to compute feature selection $F_{sel}(n)$ with concrete variable as represented in Equation (3);
- Form an encoding structure to obtain a low-dimensional hidden representation $\alpha_{enc}^{\Psi}(F_{sel})$ with the weight matrix as denoted in Equation (4);
- Perform decision making with the use of softmax layer $\zeta^h(F_{sel})$ as represented in Equation (5);
- Compute the decoder function $\alpha_{dec}^{\Psi}(F_{sel})$ with another hidden representation $\tilde{h} \in Rh'$ and defines the last reconstruction layer separately by using Equation (6);
- Obtain the selected features from the decoder result $\mathcal{S}_{\mathcal{F}}$ with the weight matrix $\varpi_{D_L}$ as shown in Equation (7);

---

B.　　**Boosted Variance Quantization Neural Network (BVQNN) Classification:** The intrusion prediction and classification are carried out using the BVQNN technique, following the selection of features using the weight matrix. Earlier studies have used a variety of machine learning and deep learning methods to create successful intrusion detection systems for protecting IoT networks. The steps of BVQNN algorithm are described in Algorithm 2. However, the main issues with the current models include longer processing times, excessive overfitting, and increasing computing complexity. Determining the type of intrusion from the provided dataset using an innovative and efficient classification algorithm known as BVQNN is the goal of the presented work. This algorithm incorporates the procedures of adaptive boost classification and quantization neural network mechanisms for accurate prediction and classification. The most popular technique is often adaptive boosting, which enables the developer to continue integrating weak learners, whose precision is merely modest until an expected minimum training error is attained. It is "adaptive" since it does not necessitate prior understanding of the veracity of these assumptions. On the contrary, it evaluates the precision of a base hypothesis after each iteration and adapts its parameters. According to a weighted training dataset, it repeatedly generates base hypotheses. In accordance with how well component hypotheses perform in classification, the weights are adaptively adjusted.

---

**Algorithm 2: Boosted Variance Quantization Neural Network**

---

Input: Training data $S_x$, test data $T_x^e$, associated distribution weight $\hat{W}_j$;

Output: Classified attack $\delta_A$;

Procedure:

Step 1: Initialize the training data $S_x$ and weight value $\hat{W}_j$ as shown in Equations (8) and (9);

Step 2: for $t = 1, \ldots, \mathbb{T}$ // $\mathbb{T}$ total number of iterations

　　　　　Train the classifier with the weighted sample $\{S_x, \hat{W}_j\}$ based on the hypothesis $H^t : \dot{x} \to [0,1]^g$;

Step 3: Estimate the Kohavi–Wolpert variance $\vartheta_{(t)}$ at the time of training by using Equation (10);

Step 4: Label probability computation $L_P^t(k)$ with the weighted label value as shown in Equation (11);

Step 5: Update the weight value $\hat{W}_j(t+1)$ according to the probability of input sample and weight vector by using Equation (12);

Step 6: Normalize the weight value $\hat{W}_j$ as shown in Equation (13);

　　　　　　　　　　end for;

Step 7: Produce the final classified output $\delta_A$ as indicated in Equation (14).

---

A weighted average is used to integrate the resulting hypotheses according to their diversity. In this technique, the training data $S_x$, testing data $T_x^e$, and associated distribution weight $\hat{W}_j$ are taken as the inputs for processing, and the classified attack $\delta_A$ is produced as the output. At first, the training data is initialized as shown in the following model:

$$S_x = \left\{ (\dot{x}_1, \dot{y}_1), (\dot{x}_2, \dot{y}_2), \ldots, (\dot{x}_{\grave{N}}, \dot{y}_{\grave{N}}) \right\} \tag{8}$$

$$\hat{W}_j = \frac{1}{\grave{N}} \ \forall \, j = 1, 2, \ldots, \grave{N} \tag{9}$$

where $\grave{N}$ is the total number of sample data, $\dot{x}$ feature data for each samples, and $\dot{y}$ is the label data for each sample. Then, the classifier training is performed with the weighted sample by using hypothesis $H^t : \dot{x} \to [0,1]^g$. Consequently, the Kohavi–Wolpert variance $\vartheta_{(t)}$ of the present training process is computed as represented in the following model:

$$\vartheta_{(t+1)} = \frac{1}{\grave{N} R_{CC}^2} * \sum_{k=1}^{\grave{N}} C\{\dot{x}_k\} * [R_{CC} - C\{\dot{x}_k\}] \tag{10}$$

where $R_{CC}$ is the number of base classifiers in ensemble technique and $C\{\dot{x}_k\}$ is the number of classifiers that correctly classifies $\dot{x}_k$. Moreover, the probability computation is per-

formed for making an effective decision while predicting the type of intrusions, which is represented in the following equation:

$$L_P^t(k) = (M-1) * \left\{ \log\left(\varpi_k^t(\dot{x})\right) - \frac{1}{M} * \sum_{i=1}^{M} \varpi_i^t(\dot{x}) \right\} \quad \forall\, k = 1, 2, \ldots, M \tag{11}$$

$$\varpi_k^t(\dot{x}) = pr^\varpi\left(H^t(\dot{x}) = k\middle|\dot{x}\right) \tag{12}$$

where $\varpi_k^t(\dot{x})$ is the weighted label probability of class k and M training vector belonging to a cluster k. Then, the weight value updation is performed with the probability of the input sample as shown in the following model:

$$\hat{W}_j(t+1) = \hat{W}_j(t) * \exp^{\left[-\frac{M-1}{M}*\log(\varpi_k^t(\dot{x}))*H^t(\dot{x})\right]} \quad \forall\, j = 1, 2, \ldots, \dot{N} \tag{13}$$

$$\varpi_k(\dot{x}) = pr(\dot{x}) \tag{14}$$

where $pr(.)$ indicates the probability of input sample with respect to weight vector. Then, the weight value normalization is performed as follows:

$$\hat{W}_j = \frac{\hat{W}_j}{\sum_{r=1}^{\dot{N}} \hat{W}_r} \quad \forall\, j = 1, 2, \ldots, \dot{N} \tag{15}$$

The final classified prediction result $\delta_A$ is represented as shown in below:

$$\delta_A = \underset{k}{argmax} \sum_{t=1}^{\mathbb{T}} \vartheta_{(t)} * L_P^t(\dot{x})_k \tag{16}$$

By using this label, the type of intrusion is exactly recognized from the dataset, and the performance evaluation is carried out to test the efficacy of predicted result.

**Theorem 1.** *Kohavi and Wolpert developed the Kohavi–Wolpert variance that is a kind of decomposition model used for estimating the error rate of the bias-variance classification approach. In this technique, the variability of the predicted label l is predicted for the sample data $\dot{x}$ data as shown in below:*

$$\sigma_{\dot{x}} = \frac{1}{2}\left(1 - \sum_{d=1}^{a_c} p(l = \tau_d|\dot{x})^2\right) \tag{17}$$

*where $a_c$ is the number of categorizations in label, l is the label vector, and $\tau_d$ is the number of samples presented in each category. According to the predicted output $a_c = 2$ , the equation $p(l = 1|\dot{x})^2 + p(l = -1|\dot{x})^2 = 1$ can be updated as follows:*

$$\sigma_{\dot{x}} = \frac{1}{2}\left(1 - p(l = 1|\dot{x})^2 - p(l = -1|\dot{x})^2\right) \tag{18}$$

$$\sigma_{\dot{x}} = p(l = 1|\dot{x}) * p(l = -1|\dot{x}) \tag{19}$$

To quantify the range of an ensemble, Kuncheva and Whitaker developed an updated version of equation as follows:

$$K_W = \frac{1}{NL^2} \sum_{i=1}^{N} C^i\left(L - C^i\right) \tag{20}$$

where $C^i$ is the number of base classifiers used for classifying the training sample $\dot{x}_i$ incorrectly, L the number of base classifiers, and N is the number of samples in $\dot{x}$. Also, as the KW variance values rise, the variety also rises.

C. **Multi-Hunting Reptile Search Optimization Algorithm (MH-RSO):** The MH-RSO algorithm is utilized in this study to forecast intrusions and classify them accurately by computing the classifier's probability value in the best possible way. Numerous optimization strategies are employed in the current studies to solve several complex problems in an appropriate manner. In the proposed study, a unique MH-RSO technique is employed to categorize the intrusions in the dataset while computing the best probability weight value. Its main advantages over conventional optimization algorithms include accelerated convergence, high efficiency, and an optimal solution with few iterations. The search for the ideal value is known as optimization, where a minimal problem is typically created by transforming the optimization problem. The boundary's lowest value, which further fulfills the constraint conditions, is the ideal global value. The reptile search algorithm models how crocodiles' approach and pursue prey. The algorithm is broken down into four stages: exploration, which includes fast crawling and belly jogging, and exploitation, which includes collaborative and cooperative hunting. Crocodiles are going to approach their prey after spotting them during the early and final phases of the hunt. Even if MH-RSO possesses some fractional optimization capabilities, it will be difficult for MH-RSO to converge on the middle and late phases of a complicated issue if MH-RSO cannot identify a rough location of the best solution in its initial and intermediate stages. The hunting behavior of MH-RSO is illustrated in Figure 2 and described in Algorithm 3.

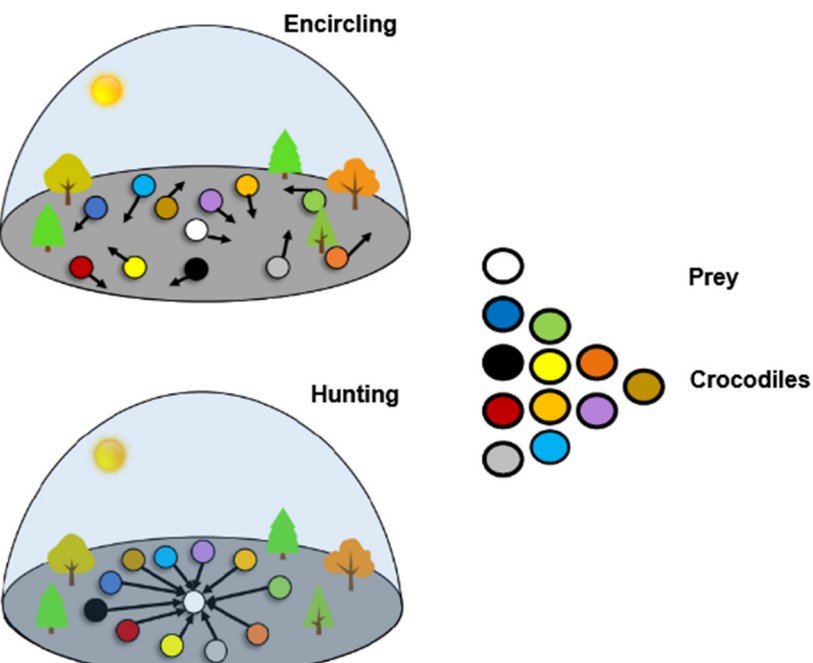

**Figure 2.** MH-RSO hunting model utilized in this paper.

This optimization algorithm includes the following modules:

- Parameter initialization;
- Encircling;
- High walking;
- Belly walking;
- Hunting.

In this technique, the feature matrix $F_x$ is taken as the input for processing, and the optimal value $best^i$ is produced as the output. After initializing the input parameters, the fitness value $Val_{fit}$ is estimated for each population as represented in below:

$$Val_{fit} = fitness(F_x) \qquad (21)$$

After that, the optimal position is estimated with the upper and lower values as shown in the following model:

$$F_x^i = L_b(i) + \mathfrak{r}*(U_b(i) - L_b(i)) \text{ where } \mathfrak{r} \in [0,1] \tag{22}$$

where $L_b(i)$ represents the lower bound, $U_b(i)$ is upper bound, and $\mathfrak{r}$ is a random number. The searching parameter is updated is by using the following model:

$$ES = 2*\mathfrak{R}*\left(1 - \frac{round}{I_{ter}}\right), \mathfrak{R} \in [-1,1] \tag{23}$$

According to the dimensionality index, the hunting operators are updated as represented as shown in the following equations.

$$o_h^i(round + 1) = best^i(round) * pr_i(round + 1) \tag{24}$$

Consequently, the probability of movement $pr_i$ is estimated as shown in below:

$$pr_i(round + 1) = \alpha + \frac{F_x^i(round) - G\left(F_x^i\right)}{best^i(round) * U_b(i) - L_b(i) + \varphi} \tag{25}$$

where $best^i$ is the best candidate population, and $\varphi$ represents the constant with positive value. Then, the round of operations are performed to find the best optimal solution, which can be used to estimate the probability value for making efficient classifications.

$$F_x^i(round + 1) = best^i(round) * \left(-o_h^i(round + 1)\right) * \beta * F_x^i(round) * \mathfrak{r} \tag{26}$$

---

**Algorithm 3: Multi-Hunting Reptile Search algorithm (MH-RSA)**

---

Input: feature matrix $F_x$;
Output: Optimal value $best^i$;
Procedure:
Step 1: Initialization parameters:
        Candidate solution R; Dimension of each solution dim; Total iteration $I_{ter}$;
        Control the search accuracy $\alpha$; Control the sensitive parameter to search capability $\beta$;
Step 2: Initialize population $F_x = \left[F_x^1, F_x^2, \ldots, F_x^n\right]$;
Step 3: While round $< I_{ter}$
        Calculate each individual's fitness value of the population as shown in Equation (21);
Step 4: Find the optimal position $F_x^i$ with lower bound $L_b$ and upper $U_b$ bound values as defined in Equation (22);
Step 5: The searching parameter ES is updated by using Equation (23);
Step 6: For each dim index by i
        By using formulas from Equations (24) and (25), the parameters $o_h$, pr, and R are updated for hunting operation;
Step 7: If round $\le I_{ter}/4$ then
        Else if round $> \frac{I_{ter}}{4}$ and round $\le I_{ter}/2$ then

$$F_x^i(round + 1) = best^i(round) * F_x^i(round) * ES * \mathfrak{r} \tag{27}$$

Else if round $\le 3*\left(\frac{I_{ter}}{4}\right)$ and round $> I_{ter}/2$ then
$$F_x^i(round + 1) = best^i(round) * pr_i(round + 1) * ES * \mathfrak{r} \tag{28}$$

Else if round $> 3*\left(\frac{I_{ter}}{4}\right)$
$$F_x^i(round + 1) = best^i(round) - o_h^i(round + 1) * \varphi - F_x^i(round) * \mathfrak{r} \tag{29}$$

        End if;
Step 8: End for; End for;
Step 9: round = round + 1;
Step 10: End while;
Step 11: Return the best solution;

---

## 4. Results

This section verifies the effectiveness and outcomes of the suggested VBQ-Net model by using a number of parameters. This work's original contribution is the creation of a cutting-edge security framework for protecting IoT networks. This study implements a combination of intelligence algorithms, including VSBW, BVQNN, and MH-RSO, for this reason. Additionally, recent benchmarking datasets, including IoT-23, IoTID-20, and CIDDS-001, which are the most prominent datasets widely utilized in many IoT security application systems, are used in the system installation and performance assessment. The VBQ-Net framework uses the VSBW technique to select relevant features, BVQNN technique to identify potential intrusions, MH-RSO to compute optimal probabilities, and evaluate performance using various metrics. The result is a more efficient and accurate analysis of performance and computational processing time.

### 4.1. Statistical Analysis

A few performance metrics are employed to verify and test the classification predictions. Typically, detection rate, accuracy, precision, recall, and f1 score are the main parameters used to validate the findings of the suggested security model. The most effective model for identifying associations among variables depends on how accurate the model is. An evaluation of the particular class is represented by the following equation:

$$\text{Detection rate} = \frac{\text{TP}}{\text{TP} + \text{FN}} \tag{30}$$

$$\text{Accuracy} = \frac{\text{TP} + \text{TN}}{\text{TP} + \text{FP} + \text{FN} + \text{TN}} \tag{31}$$

$$\text{FPR} = \frac{\text{FP}}{\text{FP} + \text{TN}} \tag{32}$$

By contrasting the actual positive estimations, it evaluates how many accurate optimistic predictions a model has made, and the precision is estimated by using the following model:

$$\text{Precision} = \frac{\text{TP}}{\text{FP} + \text{TP}} \tag{33}$$

It actually possesses a positive rate, which is a measurement of pessimistic forecasts that the model categorizes in comparison to the appropriate positive values in the actual data.

$$\text{Recall} = \frac{\text{TP}}{\text{FN} + \text{TP}} \tag{34}$$

Moreover, the f1 score is a combination and average of precision and recall, which is estimated as shown in below:

$$\text{f1} - \text{score} = 2 \times \frac{\text{Precision} \times \text{Recall}}{\text{Precision} + \text{Recall}} \tag{35}$$

where TP is true positive, TN is true negative, FP is false positive, and FN is false negative. Figure 3 shows the confusion matrix of the proposed VBQ-Net model with the predicted and actual classes.

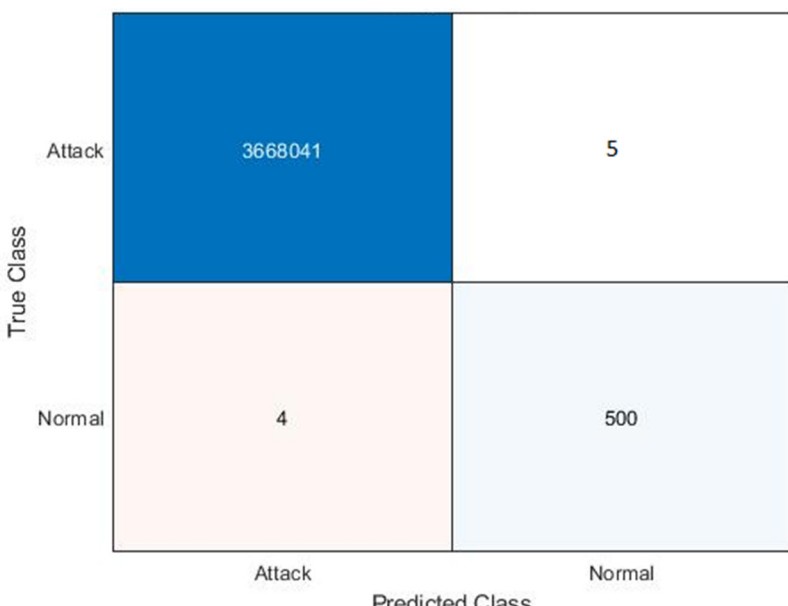

**Figure 3.** Confusion matrix for proposed system with attack and normal categories.

*4.2. Results Analysis*

The accuracy of the detection is typically used to gauge the attack detection performance and efficiency of the categorization approach. The confusion matrix, in which the classifiers categorize the real and true classes, may be more helpful in estimating the accuracy level. It is clear from the generated matrix that the suggested model successfully predicts the actual classes of attacks. Additionally, as depicted in Figure 3, the Receiver Operating Characteristics (ROC) are examined for the proposed model in relation to the TPR and FPR. The classification issues are solved using ROC and area under the curve (AUC) at various threshold values. The level, or degree of separation, is represented by the AUC. A probability curve resembles a ROC. It reveals the extent to which the model can differentiate among the classes. According to the findings, for each class of attack, the TPR of the proposed classifier is significantly raised. The proposed VBQ-Net model's ROC performance with and without feature selection models is validated in Figure 4. The obtained results show that, when combined with the VSBW approach, the suggested VBQ-Net model offers improved ROC performance outcomes. Since the suggested approach heavily relies on feature selection, it can achieve a higher attack detection rate. The VSBW is used to construct the weighted feature matrix in accordance with the encoding and decoding processes.

The training, testing, and validation accuracy of the proposed VBQ-Net model and the traditional deep learning models are shown in Figure 5. The effectiveness of the intrusion detection methodology is often assessed based on the classifier's performance during training, testing, and validation. In this study, the performance outcomes from training and testing are validated and evaluated using some of the most popular deep learning approaches, including LSTM, CNN, and hybrid CNN + LSTM [33]. Overall, the results indicate that, compared to conventional deep learning techniques, the proposed VBQ-Net offers increased training, testing, and validation accuracy of up to 99%. The training loss, testing loss, and validation loss factors are taken into consideration for evaluation, and as a result, the loss factors of the existing and suggested methods are validated and compared as shown in Figure 6. The loss factor is one of the most important parameters used to assess a classifier's performance, along with accuracy. Additionally, the results show that the VBQ-Net model outperforms the current deep learning techniques by offering lower losses of up to 4%.

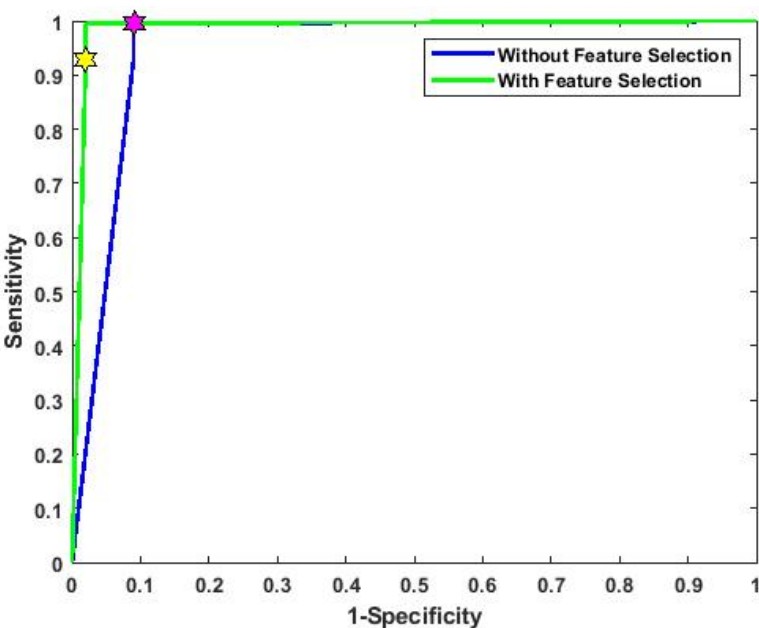

**Figure 4.** ROC analysis with and without feature selection.

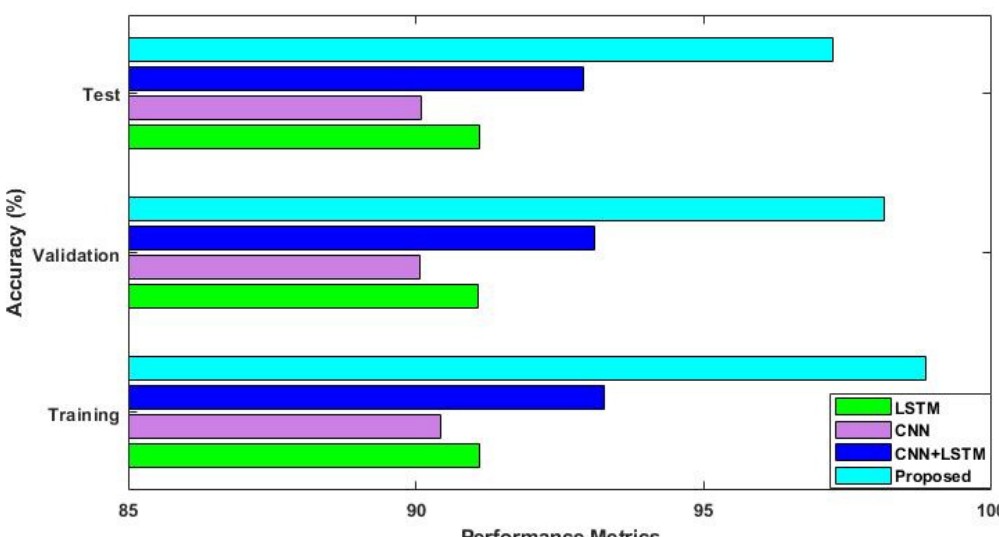

**Figure 5.** Accuracy analysis.

In Figure 7, the detection rates of the proposed VBQ-Net model and the current CNN+LSTM are compared for different types of attacks in the dataset. The detection rate, which is increased to its maximum to ensure improved performance, is used to evaluate the intrusion detection system's overall performance. The results show that, in comparison to the hybrid CNN+LSTM technique, the suggested VBQ-Net offers higher detection accuracy. Since VSBE-based feature selection and MH-RSO-based probability estimation are the major reasons for gaining an increased detection rate, the accuracy of the multi-class prediction models is validated and compared, as shown in Figure 8. By using the IoTID-20 dataset, the accuracy is evaluated and compared with several contemporary machine learning algorithms, as shown in Figure 9. These predictions show that, in comparison to both machine learning and deep learning techniques, the proposed VBQ-Net model offers the maximum accuracy.

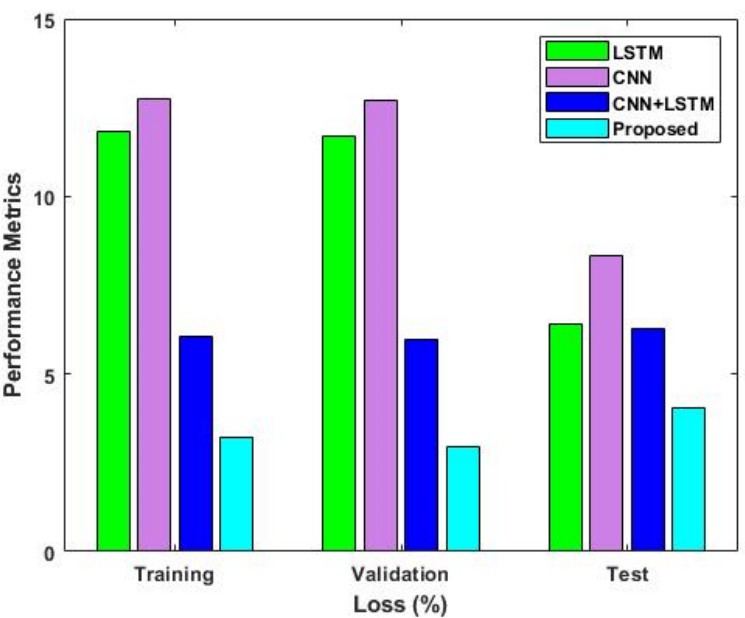

**Figure 6.** Comparative analysis based on loss.

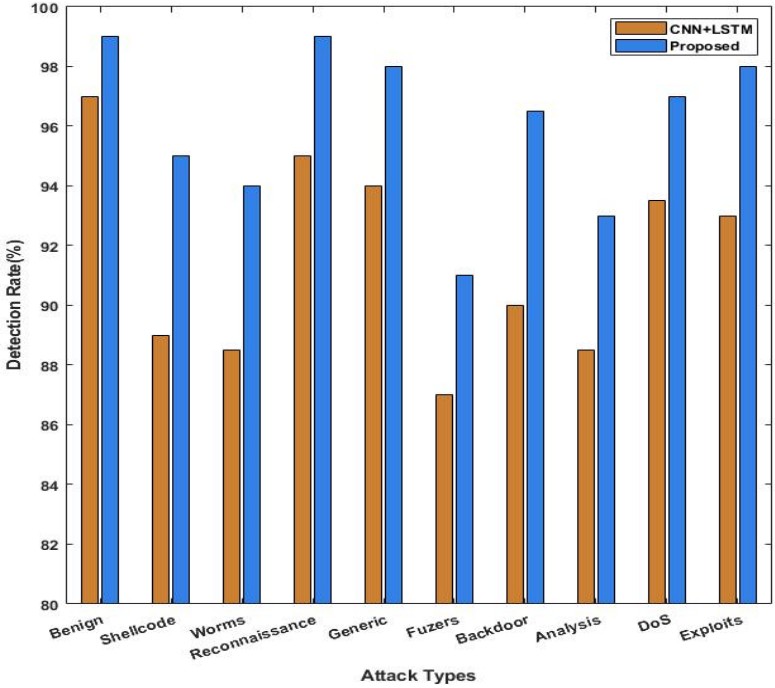

**Figure 7.** Detection rate with respect to different types of attacks.

By using the IoTID-20 dataset, Figure 10 validates and contrasts the processing times of the established and novel approaches. Usually, the processing time is calculated based on how long it took the prediction model overall to find the intrusion in the given dataset. The intrusion detection speed of the classifier is significantly boosted by the suggested framework's approach to feature selection and optimization, which significantly reduces the overall processing time. Additionally, the IoTID-20 dataset is used to validate and compare the precision, recall, and f1 score of the existing and suggested techniques, as shown in Figure 11. The results show that the VBQ-Net offers enhanced performance outcomes that are far superior to those of the traditional models.

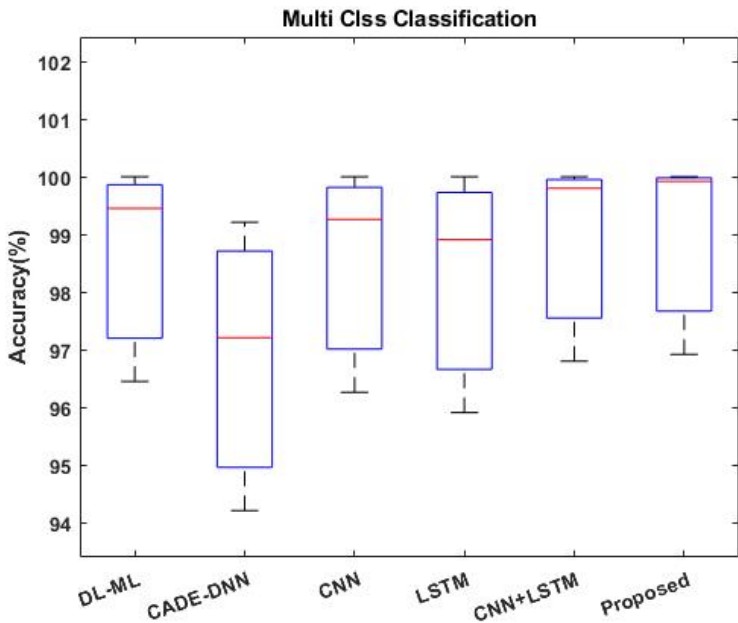

**Figure 8.** Comparative analysis with multi-class prediction models.

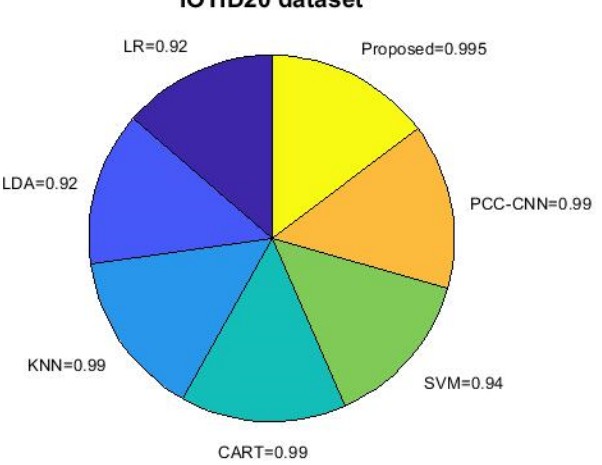

**Figure 9.** Accuracy analysis using IoTID-20 dataset.

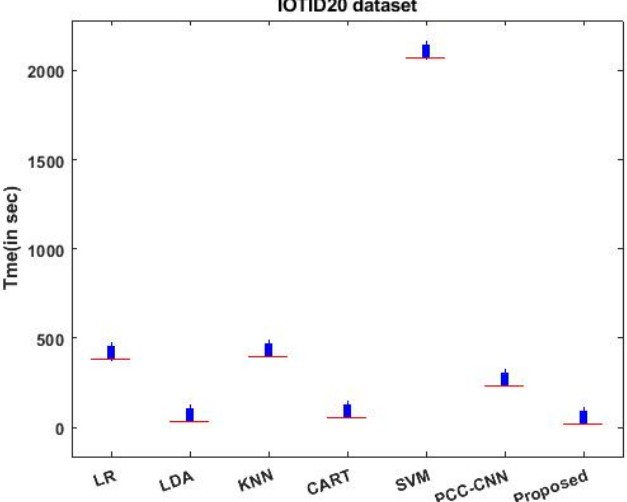

**Figure 10.** Computational processing time.

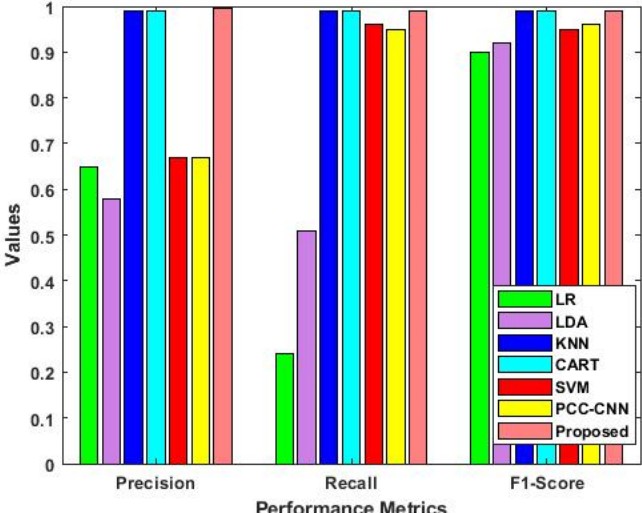

**Figure 11.** Computational processing time.

All evaluation metrics, such as precision, recall, f1 measure, accuracy, FPR, FNR, MCC, and others, are taken into account for this assessment. The dataset's dimensionality is decreased as a result of careful feature selection, increasing the classifier's accuracy. The probability weight value is calculated during classification using the MH-RSO technique's optimal solution as a base. Therefore, when compared to other conventional methodologies, the suggested VBQ-Net's overall performance results are significantly better than others.

For this evaluation, the CIDDS-001 dataset is used, and Figures 12 and 13 show that the existing approaches and suggested VBQ-Net models have the same detection rate in terms of different evaluation measures and time, respectively. The results show that, compared to the traditional model, the proposed VBQ-Net model offers a higher detection rate. The suggested framework effectively increases the attack detection rate through correct feature selection and categorization compared to others [34]. False prediction analysis on this dataset shows in Figure 14.

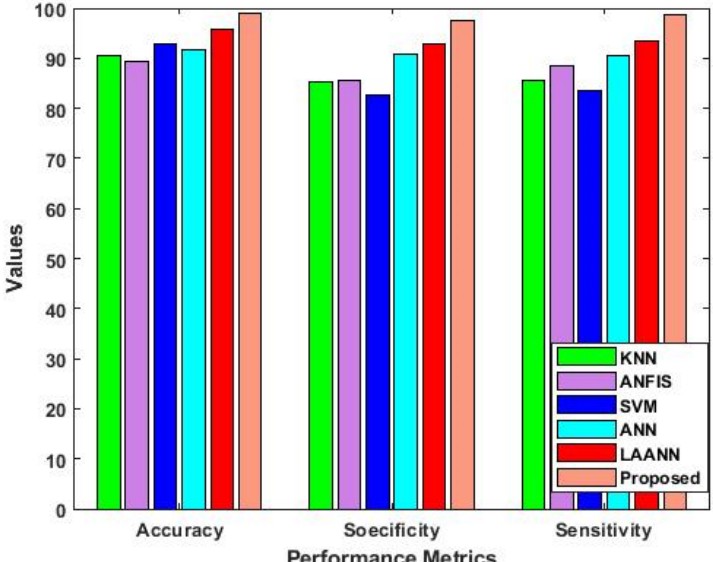

**Figure 12.** Comparative analysis using CIDDS-001 dataset.

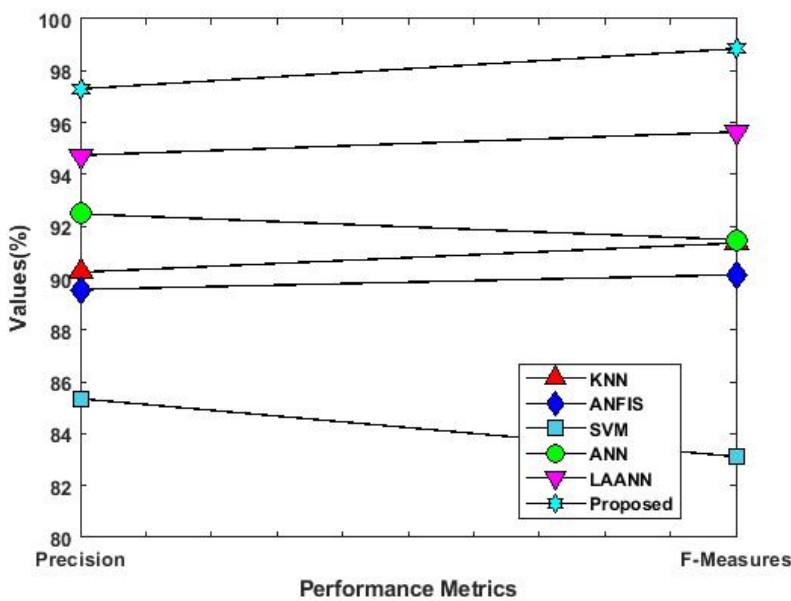

**Figure 13.** Prediction performance using CIDDS-001 dataset.

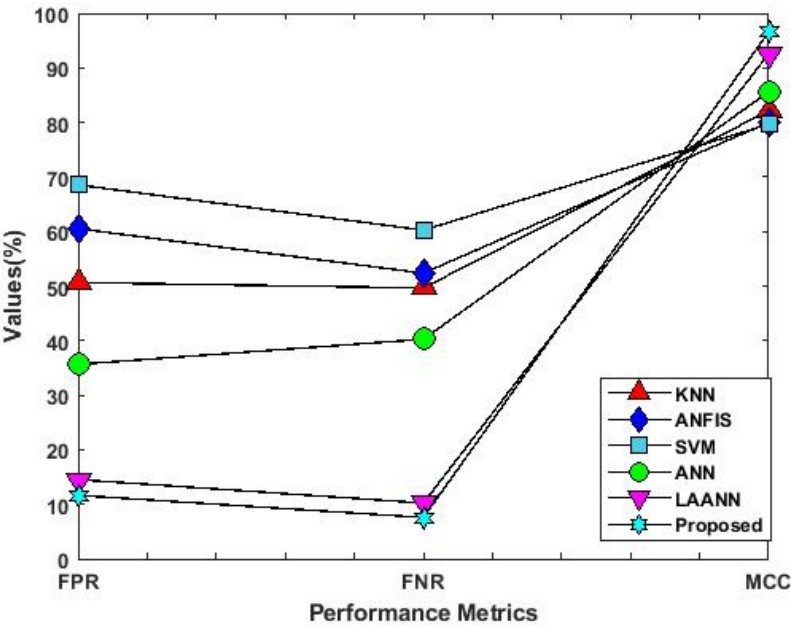

**Figure 14.** False prediction analysis using CIDDS-001 dataset.

For this evaluation, the IoT-23 dataset is used, and Figures 15 and 16 show that the existing DS-SloEL [35] and suggested VBQ-Net models have the same detection rate in terms of different evaluation measures and time, respectively. The results show that, compared to the traditional DS-SloEL model, the proposed VBQ-Net model offers a higher detection rate [35]. The suggested framework effectively increases the attack detection rate through correct feature selection and categorization.

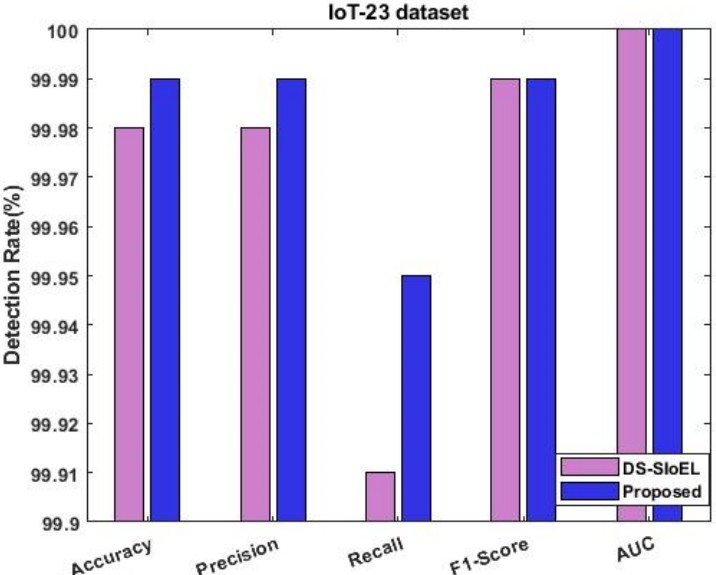

**Figure 15.** Detection rate using IoT-23 dataset.

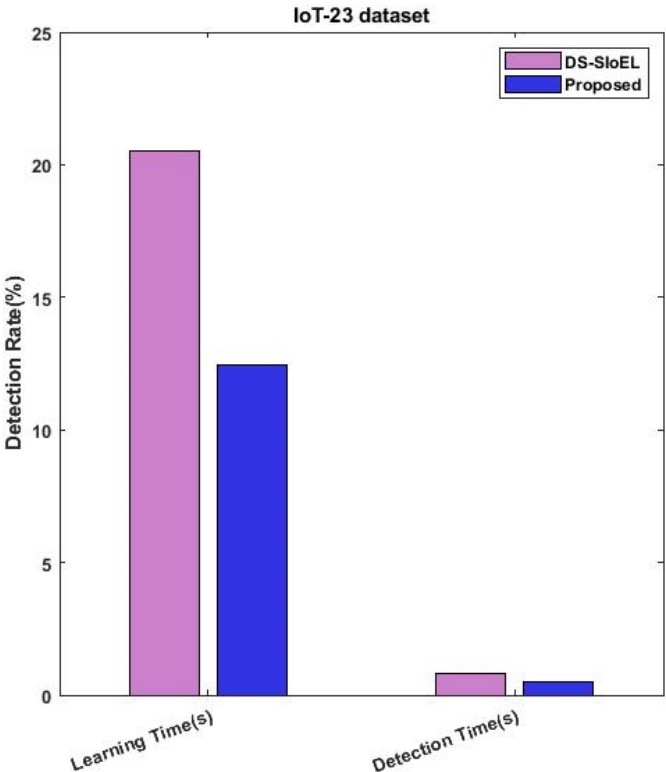

**Figure 16.** Computational time of learning and detection rate.

## 5. Discussions

The Internet of Things (IoT) has revolutionized data sharing across wireless networks, enabling seamless communication between devices. However, the growing prevalence and technological advancements of IoT systems have made them highly susceptible to cyberattacks, necessitating robust defense mechanisms [36]. This work introduces an innovative security architecture called Vectorization-Based Boost Quantized Network (VBQ-Net) to protect IoT networks. The architecture utilizes a Vector Space Bag of Words (VSBW) methodology to reduce feature dimensionality and extract key characteristics from the data. Additionally, a novel classification technique named Boosted Variance

Quantization Neural Networks (BVQNN) is employed for intrusion detection using a weighted feature matrix. To further enhance categorization, a Multi-Hunting Reptile Search Optimization (MH-RSO) algorithm is utilized to calculate probability values for identifying intrusions. Extensive validation using well-known datasets, including IoTID-20, IoT-23, and CIDDS-001, demonstrates the effectiveness of the proposed methodology.

The rise of IoT has revolutionized various industries by connecting numerous devices and enabling seamless data exchange. Yet, this interconnectedness exposes IoT systems to potential vulnerabilities and security breaches. Cyber attackers can exploit these weak points to gain unauthorized access, leading to disastrous consequences for both individuals and organizations. Traditional security measures, such as firewalls and encryption, may fall short in combating sophisticated attacks. To counteract this evolving threat landscape, our research focuses on developing an innovative security model, VBQNM.

The VBQNM leverages the power of vectorization and boosting techniques to enhance the robustness of IoT security. By transforming data into vectorized representations, the model can efficiently process and analyze incoming information, thereby enabling faster detection and response times to potential intrusions. Additionally, the utilization of boosting algorithms enables the model to dynamically adapt and improve its accuracy over time, keeping up with emerging attack strategies.

Quantization, a key component of VBQNM, plays a crucial role in reducing the computational complexity and memory requirements of the model. By quantizing the network parameters, we strike a balance between accuracy and efficiency, making VBQNM suitable for resource-constrained IoT devices without compromising security.

To evaluate the efficacy of the VBQNM, we conducted extensive experiments using real-world IoT datasets and simulated intrusion scenarios. The results demonstrate that our model outperforms conventional intrusion detection systems in terms of both accuracy and efficiency. The quantization process significantly reduces memory overhead, making it suitable for resource-limited IoT devices. Furthermore, the adaptive nature of the boosting algorithms ensures that the model continuously improves its ability to identify new and sophisticated intrusion patterns.

In this paper, we introduce a pioneering Vectorization-Based Boost Quantized Network Model (VBQNM) as a novel approach to safeguard IoT systems from hazardous intrusions. By harnessing the power of vectorization and boosting while employing quantization, VBQNM provides a comprehensive and efficient security solution for IoT environments. Our experimental results demonstrate its effectiveness in real-world scenarios, presenting a promising direction for securing the ever-expanding IoT landscape. As IoT adoption continues to grow, we believe VBQNM will play a pivotal role in ensuring the safety and privacy of IoT systems worldwide. Table 2 summarizes the state-of-the-art comparisons of machine-learning algorithms.

*Significance of Proposed System*

The current approach, Vectorization-Based Boost Quantized Network (VBQ-Net), offers several distinct advantages for safeguarding IoT networks from cyberattacks:

1. VBQ-Net leverages the power of the Vector Space Bag of Words (VSBW) methodology to effectively reduce the dimensionality of features extracted from IoT data. This process allows the model to focus on key characteristics, improving its ability to detect intrusions accurately and efficiently;
2. The novel Boosted Variance Quantization Neural Networks (BVQNN) classification technique employed in VBQ-Net enhances the accuracy of intrusion identification. By using a weighted feature matrix, the model can prioritize relevant features and classify different types of intrusions with higher precision;
3. With the quantization technique integrated into the architecture, VBQ-Net optimizes computational complexity and memory requirements. This leads to reduced latency and processing time, making it suitable for real-time intrusion detection in resource-constrained IoT environments;

4. The incorporation of the Multi-Hunting Reptile Search Optimization (MH-RSO) algorithm enables VBQ-Net to calculate probability values during intrusion categorization. This approach helps the model make more informed decisions, improving its ability to anticipate and respond to intrusions effectively;

5. VBQ-Net is designed to work with a variety of datasets, including well-known and up-to-date ones such as IoTID-20, IoT-23, and CIDDS-001. Its adaptability makes it applicable to different IoT deployment scenarios, enhancing its generalizability;

6. As an innovative security architecture, VBQ-Net strikes a balance between accuracy and efficiency. The quantization of network parameters reduces memory overhead and computational complexity, making it suitable for IoT devices with limited resources;

7. By combining multiple advanced techniques, VBQ-Net provides a comprehensive security solution for protecting IoT networks. It addresses the primary shortcomings of current IoT security frameworks, such as inadequate intrusion detection and substantial latency, making it a promising defense mechanism against cyberattacks;

8. The validation of VBQ-Net on popular and current IoT datasets demonstrates its effectiveness in real-world scenarios. Its performance in detecting intrusions and safeguarding IoT networks showcases its practical applicability in the field of IoT security.

**Table 2.** Summarizing some state-of-the-art machine learning methods used in the discussed research works for intrusion detection in IoT networks.

| Machine Learning Method | Description | Advantages | Limitations |
| --- | --- | --- | --- |
| Consistency Algorithm | Detects jamming attacks in IoT networks. | Simple implementation. | May have limitations in handling more complex attack types. |
| Deep Transfer Learning | Utilizes auto-encoders for attack identification. | Can learn complex patterns in data. | Time-consuming for detection in some cases. |
| Hybrid IDS | Combines signature-based and behavioral detection. | Effective at detecting both known and unknown attacks. | Complexity in designing and integrating different methods. |
| Kitsune-based IDS | Categorizes benign and malignant traffic in IoT networks. | Effective in handling temporal statistics of data packets. | Performance may vary depending on the dataset and features. |
| Support Vector Machine | Divides data into distinct classes for intrusion detection. | Performs well in high-dimensional source fields. | Choosing the appropriate kernel function can be challenging. |
| Logistic Regression | Used for binary classification tasks. | Simple and interpretable. | Limited performance for complex multi-class problems. |
| K-Nearest Neighbors | Classifies data based on its proximity to neighbors. | Simple and easy to implement. | Computationally intensive for large datasets. |
| Principal Component Analysis (PCA) | Reduces dimensionality of the dataset. | Helps streamline the classification process. | Information loss due to dimensionality reduction. |

In summary, the advantages of the current approach, VBQ-Net, lie in its ability to enhance intrusion detection, improve classification accuracy, reduce processing time, anticipate intrusions, optimize resource utilization, and provide robust protection to IoT networks. By addressing the limitations of existing IoT security frameworks, VBQ-Net presents a promising and innovative solution to safeguarding IoT systems from hazardous cyber intrusions.

## 6. Conclusions

Despite being a crucial part of the IoT, networks and devices have security flaws, and the majority of commonly used IoT devices have not been built with security in mind. As a result, recent attacks have been successful in infiltrating these devices and using them to carry out harmful attacks. The aim of this project is to create VBQ-Net, a new security

framework for IoT networks. The key contribution of this work is the development of a smart and effective mechanism for classifying various types of network incursions using up-to-date datasets. The framework combines innovative procedures to create a robust security system that can safeguard IoT systems. Well-known datasets such as IoTID-20, IoT-23, and CIDDS-001 have been used for system deployment and validation. The VSBW model is used to select the most important features from the dataset, with encoding and decoding procedures used for efficient feature extraction. The weighted feature matrix created in the previous stage is then used to categorize normal and incursion data using the BVQNN approach, with the MH-RSO algorithm used to make accurate decisions while anticipating data incursions. These methods have successfully detected several intrusions into the IoT network. Overall, the suggested VBQ-Net model offers improved attack detection performance, ease of understanding, rapid processing, and reduced system complexity. The training, testing, and validation accuracy, loss factor, learning and detection time, precision, recall, and other crucial performance assessment characteristics have been used in this work. The results show that, compared to traditional intrusion detection algorithms, the VBQ-Net offers superior results using the IoTID-20, IoT 23, and CIDDS-001 datasets.

While VBQ-Net is a promising approach to protecting IoT systems from harmful intrusions, it has certain limitations. The boosting algorithms used in VBQ-Net may overfit the model to the training data, especially if the model becomes too complex. To prevent this, measures such as cross-validation and regularization techniques should be employed. Highly targeted and novel intrusion techniques may still evade detection, requiring regular updates and improvements to the model. Extremely resource-constrained IoT devices may struggle to implement VBQ-Net effectively, and the model may be vulnerable to adversarial attacks. Future works should focus on evaluating its performance in diverse IoT environments, incorporating adversarial defense mechanisms, optimizing for real-time analysis, enabling adaptive learning, integrating threat intelligence, conducting comparative studies, deploying in real-world settings, ensuring model interpretability, and establishing industry-wide benchmarks.

**Author Contributions:** Conceptualization, G.P., G.S., Q.A., S.M.N. and I.Q.; Data curation, G.P., G.S., Q.A. and I.Q.; Formal analysis, G.P., Q.A. and I.Q.; Funding acquisition, Q.A. and S.M.N.; Investigation, G.S. and S.M.N.; Methodology, G.P., Q.A. and I.Q.; Project administration, Q.A., S.M.N. and I.Q.; Resources, G.P., G.S., Q.A. and I.Q.; Software, G.P., G.S., Q.A., S.M.N. and I.Q.; Validation, G.S., Q.A., S.M.N. and I.Q.; Visualization, I.Q.; Writing—original draft, G.P., G.S., Q.A., S.M.N. and I.Q.; Writing—review and editing, Q.A. All authors have read and agreed to the published version of the manuscript.

**Funding:** This work was supported and funded by the Deanship of Scientific Research at Imam Mohammad Ibn Saud Islamic University (IMSIU) (grant number IMSIU-RP23067).

**Data Availability Statement:** Not applicable.

**Acknowledgments:** This work was supported and funded by the Deanship of Scientific Research at Imam Mohammad Ibn Saud Islamic University (IMSIU) (grant number IMSIU-RP23067).

**Conflicts of Interest:** The authors declare no conflict of interest.

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
