# Peer review of "VBQ-Net: A Novel Vectorization-Based Boost Quantized Network Model for Maximizing the Security Level of IoT System to Prevent Intrusions"

_systems, doi:10.3390/systems11080436_

Round 1
Reviewer 1 Report
In this paper, the authors propose a solution that is tailored explicitly for safeguarding IoT systems from malicious intrusions. To achieve this, different types of security threats in IoT networks are detected and classified to ensure the safety of data communication. The outcome of this research is an intelligent and novel security framework, Vectorization-based Boost Quantized Network (VBQ-Net), for securing IoT systems. A number of the most recent datasets are used in this study for validating and assessing the performance of the suggested framework.
The literature review section is well-written and structured. Here, the major drawbacks behind the existing techniques have been identified. The proposed framework is developed to resolve these issues. The idea of having Table 1 is very good as summarizes the literature review.
In Section 4, you should outline the whole structure of the VBQ-Net framework. In the previous chapters, you described the used components but it should be more clear how all these are connected to form the proposed framework, i.e., in which order they used are to produce a result. The use of some figure in this case could help a lot.
In addition, both the conclusion chapter and the one above repeat similar content, which makes it look not appeal to the reader.
In line 183: All these acronyms should be explained what they mean.
In line 482: There is an acronym MH-RSO, is this a typo or is something different than the MH-RSA?
Author Response
Original Manuscript ID: ID: systems-2557588
Original Article Title VBQ-Net: A Novel Vectorization-based Boost Quantized Net-work Model for Maximizing Security-level of IoT System from Intrusions
To: Editor in Chief,
MDPI, Systems
Re: Response to reviewers
Dear Editor,
Many thanks for insightful comments and suggestions of the referees. Thank you for allowing a resubmission of our manuscript, with an opportunity to address the reviewers’ comments.
We are uploading (a) our point-by-point response to the comments (below) (response to reviewers), (b) an updated manuscript with yellow highlighting indicating changes, and (c) a clean updated manuscript without highlights (PDF main document).
By following reviewers’ comments, we made substantial modifications in our paper to improve its clarity, English and readability. In our revised paper, we represent the improved manuscript such as:
(1) Revised Abstract, (2) Revised Introduction, (3) Results section, (4) Discussions and Conclusion sections.
We have made the following modifications as desired by the reviewers:
Best regards,
Corresponding Author,
Dr. Qaisar Abbas (On behalf of authors),
Professor.

Reviewer 2 Report
Data sharing across wireless networks is simplified by the Internet of Things (IoT), but the growth and advancements of IoT systems make them vulnerable to cyberattacks. To address the shortcomings of current IoT security frameworks, this study proposes a novel security architecture called VBQ-Net, which utilizes VSBW for feature reduction, BVQNN for intrusion classification, and MH-RSO for intrusion anticipation. The effectiveness of the proposed methodology is evaluated using well-known IoT datasets, aiming to provide a more robust defense against cyberattacks on IoT systems. However the study required revisions such as:
- Add a new row to Table 1 that summarizes your study and compares it to previous studies. You can also include a checklist of features to highlight the contribution of your study.
- If there are any copyright issues with the icons or visuals in Figure 1, make sure to declare them. Additionally, consider updating Figure 1 if it appears complicated. It would be helpful to provide a clear explanation of the network features data.
- Since Bag-of-Words is used, it can be assumed that this work involves text data. Did you apply any natural language processing or text pre-processing techniques before extracting the features? Clarify this in your study.
- Provide a more detailed description of the dataset used. Include information such as the number of instances, what they are, and where you obtained the dataset.
- Clearly explain how the training process is set up, including the percentage of data used for training and testing, as well as the experimental settings. This information is crucial for reproducibility. Consider sharing the source code of the proposed model with the public.
- Figure 4-11 seems complicated and difficult to understand. Consider using a table instead, presenting the dataset used and the performance metrics. This will make it easier for readers to see how the proposed model outperformed other models.
- If a public dataset was used, it would be beneficial to include a comparison study from previous results to demonstrate the major contribution of your work.
Moderate editing of English language required
Author Response

(The authors gave the same response as above.)

Round 2
Reviewer 2 Report
I approve the comments and thanks.
I approve the comments and thanks.